# Development of a Marmoset Apparatus for Automated Pulling to study cooperative behaviors

Olivia C Meisner[1,2,3], Weikang Shi[2,3,4], Nicholas A Fagan[2], Joel Greenwood[3,5], Monika P Jadi[1,3,4,6†], Anirvan S Nandy[1,2,3,4,5*†], Steve WC Chang[1,2,3,4,5*†]

[1]Interdepartmental Neuroscience Program, Yale University, New Haven, United States; [2]Department of Psychology, Yale University, New Haven, United States; [3]Department of Neuroscience, Yale University, New Haven, United States; [4]Wu Tsai Institute, Yale University, New Haven, United States; [5]Kavli Institute for Neuroscience, Yale University School of Medicine, New Haven, United States; [6]Department of Psychiatry, Yale University, New Haven, United States

## eLife Assessment

This **valuable** study describes an apparatus, workflow, and proof-of-concept data for a system to study social cooperation in marmosets, an increasingly popular primate model for neuroscience. The apparatus and methodology have clear and **convincing** advantages over conventional methods based on manual approaches. However, claims of faster social learning or of finer-grained behavioural analysis in this setup will require further corroboration.

**\*For correspondence:**
anirvan.nandy@yale.edu (ASN);
steve.chang@yale.edu (SWCC)

†These authors contributed equally to this work

**Competing interest:** The authors declare that no competing interests exist.

**Abstract** In recent years, the field of neuroscience has increasingly recognized the importance of studying animal behaviors in naturalistic environments to gain deeper insights into ethologically relevant behavioral processes and neural mechanisms. The common marmoset (*Callithrix jacchus*), due to its small size, prosocial nature, and genetic proximity to humans, has emerged as a pivotal model toward this effort. However, traditional research methodologies often fail to fully capture the nuances of marmoset social interactions and cooperative behaviors. To address this critical gap, we developed the Marmoset Apparatus for Automated Pulling (MarmoAAP), a novel behavioral apparatus designed for studying cooperative behaviors in common marmosets. MarmoAAP addresses the limitations of traditional behavioral research methods by enabling high-throughput, detailed behavior outputs that can be integrated with video and audio recordings, allowing for more nuanced and comprehensive analyses even in a naturalistic setting. We also highlight the flexibility of MarmoAAP in task parameter manipulation which accommodates a wide range of behaviors and individual animal capabilities. Furthermore, MarmoAAP provides a platform to perform investigations of neural activity underlying naturalistic social behaviors. MarmoAAP is a versatile and robust tool for advancing our understanding of primate behavior and related cognitive processes. This new apparatus bridges the gap between ethologically relevant animal behavior studies and neural investigations, paving the way for future research in cognitive and social neuroscience using marmosets as a model organism.

## Introduction

The study of animal behavior is essential for comprehending the intricacies of behavioral dynamics and their underlying cognitive processes. Exploring the neurobiological foundations of animal behavior

**eLife digest** Cooperation is one of the most important and advanced forms of social behaviour, yet studying it in laboratory settings can be particularly challenging. This is partly because animal species typically used in research do not cooperate in a way similar to humans.

More recently, marmosets have gained recognition as an important model for studying collaboration, as these small primates naturally exhibit cooperative behaviours. However traditional research methods have struggled to capture these dynamics in a reliable and detailed way. A lack of approaches that allow researchers to methodically prompt naturalistic behaviours in freely moving animals under various controlled circumstances has hampered efforts to study the factors that influence cooperation. This limitation has also hindered investigations into the brain processes that underpin this unique social trait.

To address this gap, Meisner et al. developed MarmoAAP, an apparatus that allows two marmosets in adjacent, transparent enclosures to observe each other and coordinate their actions so they can simultaneously pull levers and both receive a reward. This tool is compatible with advanced tracking technologies to monitor behaviour and brain activity.

Testing revealed that the marmosets exhibited cooperative behaviour much more consistently and in greater numbers with MarmoAAP than in previous experiments using traditional, non-automated methods, making the apparatus an effective tool for studying this complex social behaviour.

In addition to studying cooperation, MarmoAAP offers a standardised platform for testing the effects of drugs in marmosets, which could help develop new treatments for further testing in humans. Importantly, performance on the task could be precisely quantified using the detailed metrics provided by the apparatus. This is crucial for better understanding the factors that influence cooperative ability, and how these behaviours can be enhanced or disrupted. Neuroscientists could also use this combination of adaptable design and high-resolution data gathering to better understand brain activity in a wide range of complex primate behaviours.

enables the identification of shared neural mechanisms governing decision-making, learning, memory, and problem-solving throughout the broader spectrum of the animal kingdom. However, investigations of the neurobiology of ecologically valid behaviors can be extremely challenging using traditional approaches. With the rapid advancements in methods for recording and manipulating neural activity from these species, there arises a critical need to modernize our approaches to studying animal behavior to ensure they keep pace with the evolving neural techniques (*Miller et al., 2022*; *Huk et al., 2018*; *Scott and Bourne, 2022*).

There is growing recognition of the common marmoset's (*Callithrix jacchus*) potential as an invaluable animal model in neuroscience research (*Miller et al., 2016*; *Burkart and Finkenwirth, 2015*) as evidenced by efforts to create marmoset brain databases at multiple biological levels (*Lin et al., 2019*; *Liu et al., 2020*; *Woodward et al., 2018*; *Okano et al., 2016*). Marmosets provide notable advantages as research models, including their immediate relevance to humans given their genetic relatedness and shared dominant sensory modalities (*Miller et al., 2016*; *Mitchell and Leopold, 2015*), and their small size which facilitates naturalistic, freely moving studies of primate social behaviors that can be challenging with larger species like macaques. Further, they offer a distinctive platform for the study of social behaviors due to their significant parallels with human social structures. Marmosets are particularly prosocial and socially tolerant primates that, like humans, engage in pair bonding and cooperative breeding (*De la Fuente et al., 2022*; *Schaffner and Caine, 2000*; *French, 1997*; *French et al., 2018*), which has been theorized to have significantly shaped socio-cognitive abilities. Indeed, marmosets consistently show more socio-cognitively advanced behaviors such as social learning, vocal communication, understanding and use of gaze cues, and cooperative problem-solving relative to non-cooperatively breeding primates (*Cronin et al., 2005*; *Burkart and Heschl, 2007*; *Hare et al., 2003*; *Burkart and van Schaik, 2010*; *Snowdon and Cronin, 2007*). As primates, marmosets also share significant similarities to humans in their neural circuits involved in social cognition (*Miller et al., 2016*). For example, both humans and marmosets show similar face-responsive brain regions in the temporal lobe (*Tsao et al., 2008*; *Hung et al., 2015*) and similar brain networks comprising the social brain (*Deen et al., 2023*; *Cléry et al., 2021*). Marmosets provide a unique opportunity to investigate

social behavioral dynamics, however, being a relatively new model in the field of neuroscience, they have yet to benefit from the extensive methodological developments available for other model organisms like rodents. Continued innovation in research methods is essential to fully utilize marmosets as a model system to study complex behaviors and their neural correlates.

Within the realm of animal behavior studies, investigating the dynamics of social interactions and decision-making presents both a challenge and a promising avenue for investigating complex cognitive processes. Advanced social cognition demands constructing and flexibly updating internal models of social agents and computing multiple layers of information across self and others (*Fehr and Fischbacher, 2003*; *Rilling and Sanfey, 2011*). In particular, cooperation, a key behavioral strategy crucial to the evolution of advanced social cognition, involves integrating complex information like social relationships and the goals and intentions of oneself and others (*Brosnan et al., 2010*; *de Waal, 2008*; *Nowak and Sigmund, 2005*; *Lozano et al., 2020*; *Bliege Bird et al., 2018*; *Mustoe et al., 2016*; *Fehr and Rockenbach, 2004*; *Axelrod and Hamilton, 1981*; *Boyd and Richerson, 1985*; *Brosnan, 2011*). Given the theorized role of cooperation in the evolution of higher-order cognitive processes involving processing and engaging in social interactions, studying this behavior can have important implications for understanding social dynamics, communication, and cognition in the animal kingdom (*Burkart and van Schaik, 2010*; *Burkart et al., 2014*; *Clutton-Brock, 2009*).

Traditionally, researchers have studied cooperative behaviors using the cooperative pulling paradigm, a widely employed experimental setup in several animal species that requires animals to collaborate to manipulate a device and retrieve a food reward (*Crawford, 1937*). This paradigm involves two animals working in tandem, each pulling one end of a rope looped through rings attached to a heavy board on the ground. Because the food board is either too heavy for one animal to move alone or rigged so that one animal pulling the rope does not move the board, only coordinated actions can lead to successful food acquisition. This well-established paradigm has greatly contributed to our understanding of cooperative abilities across diverse species, including, but not limited to, chimpanzees, capuchins, hyenas, wolves, dogs, elephants, otters, and rooks (*Cronin et al., 2005*; *Crawford, 1937*; *Martin et al., 2021*; *Mendres, 2000*; *Plotnik et al., 2011*; *Range et al., 2019*; *Drea and Carter, 2009*; *Seed et al., 2008*; *Schmelz et al., 2017*).

While the cooperative pulling paradigm has been invaluable in shedding light on cooperative behaviors across a variety of species, it presents several limitations that hinder its utility for investigating complex behavioral dynamics and preclude studies of underlying neural mechanisms. One notable limitation is the relatively low resolution of behavioral output variables typically measured in the traditional pulling paradigm. Researchers often categorize outcomes in terms of broad categories of successful or unsuccessful cooperation attempts, which may not capture the nuances of behavior with the precision required for advanced analyses. Moreover, manual coding of animal behaviors within the task is constrained to second-by-second measurements and necessitates substantial human labor and expertise.

Additionally, the traditional cooperative pulling paradigm requires frequent experimenter intervention to reset the apparatus's position and reload it with food rewards between trials. This time-consuming process not only disrupts the natural behaviors of the animals but also limits the number of trials that can be conducted in each session, often allowing for only a meager sample size. For example, from a sample of five experiments employing a traditional cooperative pulling task across a range of species, animals performed, on average, 10.4 trials per session (*Cronin et al., 2005*; *Martin et al., 2021*; *Mendres, 2000*; *Plotnik et al., 2011*; *Range et al., 2019*; *Drea and Carter, 2009*; *Seed et al., 2008*), which poses a significant challenge for neural investigations. Furthermore, the manipulability of task variables is constrained in this manual setup, hampering researchers' ability to investigate if and how specific factors influence cooperative behaviors in a controlled manner.

Recent studies have explored alternative cooperative tasks to address some of these limitations, including the work by *Jiang et al., 2021*, which demonstrated the potential of using a nose-poking task to study cooperation in mice, rats, and tree shrews. This study highlighted the comparative abilities of cooperation across different mammalian species and provided a framework for developing more precise and controlled cooperative tasks. Building upon this work, we aimed to develop a task that not only captures the nuances of cooperative behaviors but also allows for high-resolution data collection, making it more suitable for investigating the underlying neural mechanisms in primates.

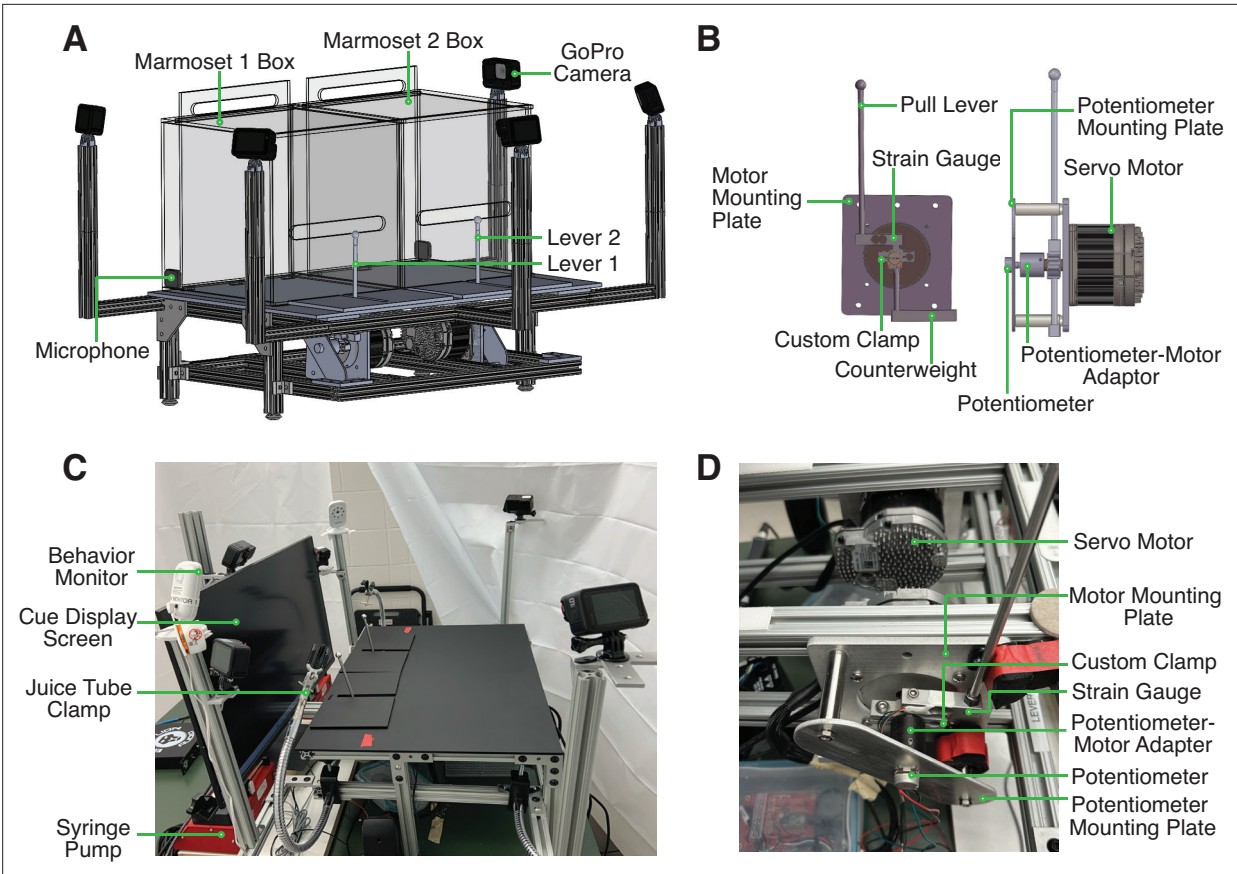

**Figure 1.** Design and Structure of Marmoset Apparatus for Automated Pulling (MarmoAAP). (**A**) CAD model of apparatus highlighting general apparatus layout, transparent testing boxes, frame structure, video cameras, and microphones. (**B**) CAD model of lever-motor assembly. Left: Side view of lever-motor assembly. Potentiometer-motor adaptor has been omitted in this view due to its obstruction of view of other components. Right: Front view of lever-motor assembly. (**C**) Photo of actual apparatus. (**D**) Photo of lever-motor assembly.

Here, we introduce an innovative method for studying cooperative behaviors in common marmosets (*C. jacchus*) using an automated apparatus and task. Our Marmoset Apparatus for Automated Pulling (MarmoAAP) for studying cooperative behaviors comprises response levers controlled by a precision servo motor, integrated with a suite of custom-designed components and sensors. Notably, MarmoAAP offers immense versatility, as it can be seamlessly programmed to accommodate a wide spectrum of behavioral tasks. We demonstrate its utility in the investigation of cooperative behaviors, highlighting its capacity to elicit a large number of trials within a single session while producing exceptionally granular behavioral readouts suitable for sophisticated analytical approaches. This not only surmounts the constraints of traditional methodologies but also aligns with the advanced tools now available for scientific research.

## Materials and methods
### Apparatus design and construction
We developed an apparatus for an automated version of the cooperative pulling paradigm for marmosets (*Figure 1*). In this setup, two marmosets are each placed in a separate transparent behavior box set atop the behavioral rig. The marmosets are able to freely move within their individual boxes and have full access to their pull levers in front of them. They can reach out of a slot in the transparent behavior boxes to pull their levers at any time. The levers are controlled by an assembly of components that allow us to program a rotational force to be exerted on the levers. This force can be used to change the position of the levers or adjust the difficulty of pulling the levers. With the incorporated sensors, we can also measure the position of and force exerted on the lever with millisecond precision.

**Table 1.** Apparatus parts and information.

| Apparatus item | Part | Manufacturer/part number | Function |
|---|---|---|---|
| Pull Lever Head | Ball Head | Custom | Lever grip suitable for marmosets |
| Pull Lever Shaft | 6-inch rod | ThorLabs / ER6-P4 | Holds lever head at appropriate height |
| Lever Bumpers | Load-Rated Threaded Bumper | McMaster-Carr/93115K121 | Bumper for lever to rest on when in starting position |
| Lever Assembly Motor | Servo Motor | ClearPath Integrated Servo System by Teknic/CPM-MCVC-3411S-RLN | Exerts rotational force on the lever |
| Motor Mounting Plate | Custom-designed aluminum mounting plate | Custom designed. Aluminum cut by water jet | Holds servo motor to base frame |
| Potentiometer | Rotary Potentiometer—10k Ohm, Linear | SparkFun Electronics/COM-09939 | Delivers real-time positional output of pull lever |
| Strain Gauge | Load cell | SparkFun Electronics/SEN-14729 | Delivers real-time force reading of pull lever |
| Potentiometer-Motor Adaptor | Custom designed | Custom designed. Aluminum cut by water jet | Yokes potentiometer shaft and motor shaft |
| Potentiometer Mounting Plate | Custom-designed aluminum mounting plate | Custom designed. Aluminum cut by water jet | Holds the potentiometer in position adjacent to the servo motor |
| Potentiometer Plate Crossbars | Ø1/2" Optical Post, SS, 8–32 Setscrew, 1/4"-20 Tap, L=2" | ThorLabs / TR2 | Fixes potentiometer mounting plate to motor mounting plate |
| Counterweight | Custom-designed aluminum bar | Custom-designed aluminum bar cut by water jet | Offsets pull lever weight to balance system |
| Control board | Teensy 3.2 USB Development Board | SparkFun Electronics/DEV-13736 | Microcontroller used to integrate motors, potentiometer, and strain gauge |
| Breadboard | Solder-able Breadboard | SparkFun Electronics/PRT-12070 | Connects Teensy microcontroller to sensors, facilitating control and computer integration |
| Syringe Pump | Syringe Pump | New Era/DUAL-NE-1000X | Syringe pump controlled by task code to deliver juice reward |
| Frame | T-slotted rails | McMaster-Carr/47065T553 | Provide a scaffolding to support lever-motor assembly |
| Cameras | Go-Pro Cameras | Go-Pro/HERO10 | Record videos of marmosets performing the task |
| Audio Recorder | Voice Recorder, 16 GB | QZTELECTRONIC (via Amazon) | Record vocalizations from marmosets performing the task |

Additionally, there are five GoPro cameras and two microphones to capture video and audio recordings, respectively.

The base frame of this apparatus is constructed of modular T-slotted framing and connectors (*Figure 1A and C*). The main structure of the base is a table measuring 18" h × 24" w × 18" l. The table consists of four legs supporting two sets of horizontal rectangular framing. The rectangular frames have T-slotted rails running from front to back, and the lever assembly is attached to these rails. T-slotted framing was also used to provide arms extension to hold the GoPro cameras in the appropriate position (*Figure 1A and C*).

The core of the apparatus is the assembly that controls the marmosets' pulling levers (*Figure 1B*). This assembly is constructed from the materials listed in *Table 1*. The movement of and force required to pull each pull lever is controlled by a servo motor that is programmed via microcontroller development boards (Teensy). With custom code (Arduino), we can exert rotational force on the levers via motor control. This enables us to change the force with which marmosets must pull the levers. It also allows us to reset the levers to the starting positions after the marmosets have pulled the levers. The levers are also connected to two sensors, a strain gauge and a potentiometer. The strain gauge converts the force exerted onto the lever into an electrical signal. The potentiometer measures the position of the lever in a rotary motion and converts it into an electrical signal. These signals can be transmitted to a computer via the Teensy board and used in the task code to evaluate the marmosets'

lever-pulling actions and contingently trigger reward delivery from the syringe pumps (New Era/DUAL-NE-1000X). A successful lever pull is determined by the lever position passing a specified positional threshold as determined by the potentiometer reading. For the Self-Reward condition, a lever pull is considered successful when either lever passes the positional threshold at any time in the session. For the Mutual Cooperation condition, a pair of lever pulls is considered successful when the second lever pull has passed the positional threshold within 1 s of the partner's lever having passed the positional threshold.

The lever-motor assembly also consists of structural components that hold each component in the appropriate position. The servo motor is attached to the base frame with a custom-designed mounting plate. A custom-designed clamp sits around the shaft of the motor and serves to yoke the strain gauge and pull lever to the movement of the motor shaft. The top side of the strain gauge connects to the top side of the clamp, and the lever is then connected to the opposite end of the strain gauge. This enables lever force reading by the strain gauge each time the lever is pulled. The bottom side of this clamp is attached to a counterweight (*Figure 1B*). Positioned directly opposite to the motor shaft is the potentiometer (*Figure 1B*). The potentiometer is yoked to the motor shaft such that when the lever moves, and therefore the motor shaft rotates, the potentiometer shaft also rotates. This ensures that the potentiometer shaft movement and therefore the potentiometer readings correspond to lever movement. To achieve this, we use a 3D-printed potentiometer-motor adapter. One side of this adapter fits onto the motor shaft and the other side fits onto the potentiometer shaft. Both shafts are held securely in place with set screws.

Finally, the body of the potentiometer must be held in a stable position so that it does not also move when the potentiometer shaft rotates. To achieve this, we designed a potentiometer mounting plate. This plate has two holes such that crossbars can be attached to connect this potentiometer mounting plate to the motor mounting plate. Additionally, it has a hole that the potentiometer shaft is passed through, and a smaller hole that the tab on the potentiometer body can be placed in. This tab holds the potentiometer in a fixed position to prevent the potentiometer body from rotating when the potentiometer shaft rotates.

## Animals

We trained a total of seven adult common marmosets (*C. jacchus*) (3 males, 4 females; 6.0±1.7 years, mean ±s.d.) to perform the MarmoAAP lever-pulling tasks. All marmosets were either pair- or group-housed and lived in the same colony room with a 12 hr light-dark cycle. All pairs tested together were familiar cage-mates. Before testing sessions, water access was temporarily removed, and the AM feed was withheld for 1–3 hr. Water and food were given upon return to the home cage after testing at which point animals had unrestricted access to both. All procedures were approved by the Yale Institutional Animal Care and Use Committee (Yale University IACUC Protocol #2023-20163) and complied with the National Institutes of Health Guide for the Care and Use of Laboratory Animals.

## Behavioral training

Marmosets were first trained to voluntarily enter a transport box. At this stage, they were also trained to target touch the metal rod used for the apparatus' lever from their home cage in exchange for a marshmallow or mealworm reward. Once marmosets were comfortable entering the transport box, they were habituated to transportation to the testing room and sitting inside the transport box in the room. During training, marmosets were always transported and habituated in cage-mate pairs. Once comfortable in the testing room, marmosets were habituated to the transparent behavior boxes (*Figure 1A*) and trained to pull the levers (*Figure 1A*) on the apparatus in exchange for a liquid reward (marshmallow fluff diluted with water; 6 g marshmallow fluff per 20 ml water).

Next, marmosets were trained to perform the Self-Reward task. For this task, marmoset pairs were placed in their separate transparent behavior boxes side-by-side, and each was free to pull their lever at any time in exchange for 0.1 ml of liquid reward. The pull-reward contingency was fully independent across the two marmosets. A monitor in front of them depicted a white square cue for this task (*Figure 1C*). Once they reliably performed the Self-Reward task, we began training them to perform the Mutual Cooperation task. For this task, we introduced a contingency requiring that they pull their levers within a certain time window of one another to receive mutual liquid rewards. A yellow circle cue was depicted on the monitor in front of them for this task. Training advanced through incremental

decreases in the cooperative time window including 3 s, 2 s, then 1.5 s, and finally 1 s. A pair of lever pulls in this condition was deemed successful cooperation if the levers were both pulled past their position thresholds within the cooperative time window. After the follower pulled their lever on a successful pull, a tone was played immediately and 0.2 ml of liquid reward was delivered to both animals 1 s later.

### Multi-animal 3D tracking

We used DeepLabCut2 (DLC2) (*Mathis et al., 2018*; *Nath et al., 2019*; *Lauer et al., 2022*) to track the head frames of marmosets. We labeled the six facial parts for both animals to define the head frames – two ear tufts, two eyes, central blaze, and mouth. The training dataset contained 270 video frames taken from three cameras in three sessions. We used the multi-animal version of the DLC2 model (*Lauer et al., 2022*), and trained the model with labeled frames from all three cameras for 15,000 iterations until the errors from the loss function reached the plateau (loss <0.001). We applied this trained model to videos taken from all three cameras. This model was also generalizable across sessions and different marmoset individuals. We use Anipose to create the 3D reconstruction of the marmosets' head frames based on videos taken simultaneously from the three cameras (*Karashchuk et al., 2021*). We first used the checkerboard method to calibrate the three cameras using Anipose, and then provided the DLC2 tracking results from all three cameras at the same time to Anipose to finalize the triangulations.

### Head chamber implantation and craniotomy

One animal received a head chamber implant and craniotomy. After the head chamber was surgically implanted, the animal was allowed to recover for 2 weeks. After the recovery, a second procedure was performed to create a craniotomy and mount a screw microdrive ('nanodrive'; Cambridge Neurotechnologies Inc) holding a 64-channel linear array electrode (NeuroNexus) onto the skull of the marmoset. Craniotomy placement was guided by CT scans and stereotaxic coordinates. The electrode's electronic interface board was then connected to the White Matter eCube headstage chips (White Matter LLC) which were secured in the marmoset's head chamber. The implanted electrode was then lowered into the desired cortical site.

### Neural recordings

Recordings were logged using White Matter's eCube headstage system. At the beginning of each recording session, the marmoset was restrained, but not head-fixed, in a chair, and the White Matter's data logger was connected to the headstage chips in the head chamber. The logger was secured in place with a cap. The marmoset was previously habituated to this restraint process, and the process typically lasted approximately 5 min. The marmoset was then transferred to their transparent behavior box, which was placed on the rig next to their partner's box, allowing both to engage in the behavioral task. On neural recording days, behavioral sessions consisted of one block of the Mutual Cooperation task and one block of the Self-Reward task, each lasting approximately 10 min. After the behavioral testing, the marmoset was again placed in the chair for removal of the data logger. Electrical signals were collected at 20 kHz from the probe. Action potential waveforms were extracted using Kilosort2 (*Pachitariu et al., 2023*) and manually sorted into single units and multi-units using phy, an open-source Python library for manual clustering of electrophysiology data.

## Results
### Marmosets perform high number of trials on automated lever-pulling tasks

The implementation of MarmoAAP yielded significant advancements in the ability to study complex behaviors in marmoset monkeys. In the initial phase of our study, we successfully trained a cohort of seven marmosets to perform the Self-Reward condition in which they could pull their lever at any time to earn a 0.1 ml juice reward for themselves. On average, marmosets pulled 163±56 (mean ± s.e.m.) times per 20 min behavioral session demonstrating high levels of motivated behavior (*Figure 2A*). This training phase thus demonstrated the marmosets' capacity to acquire and consistently execute

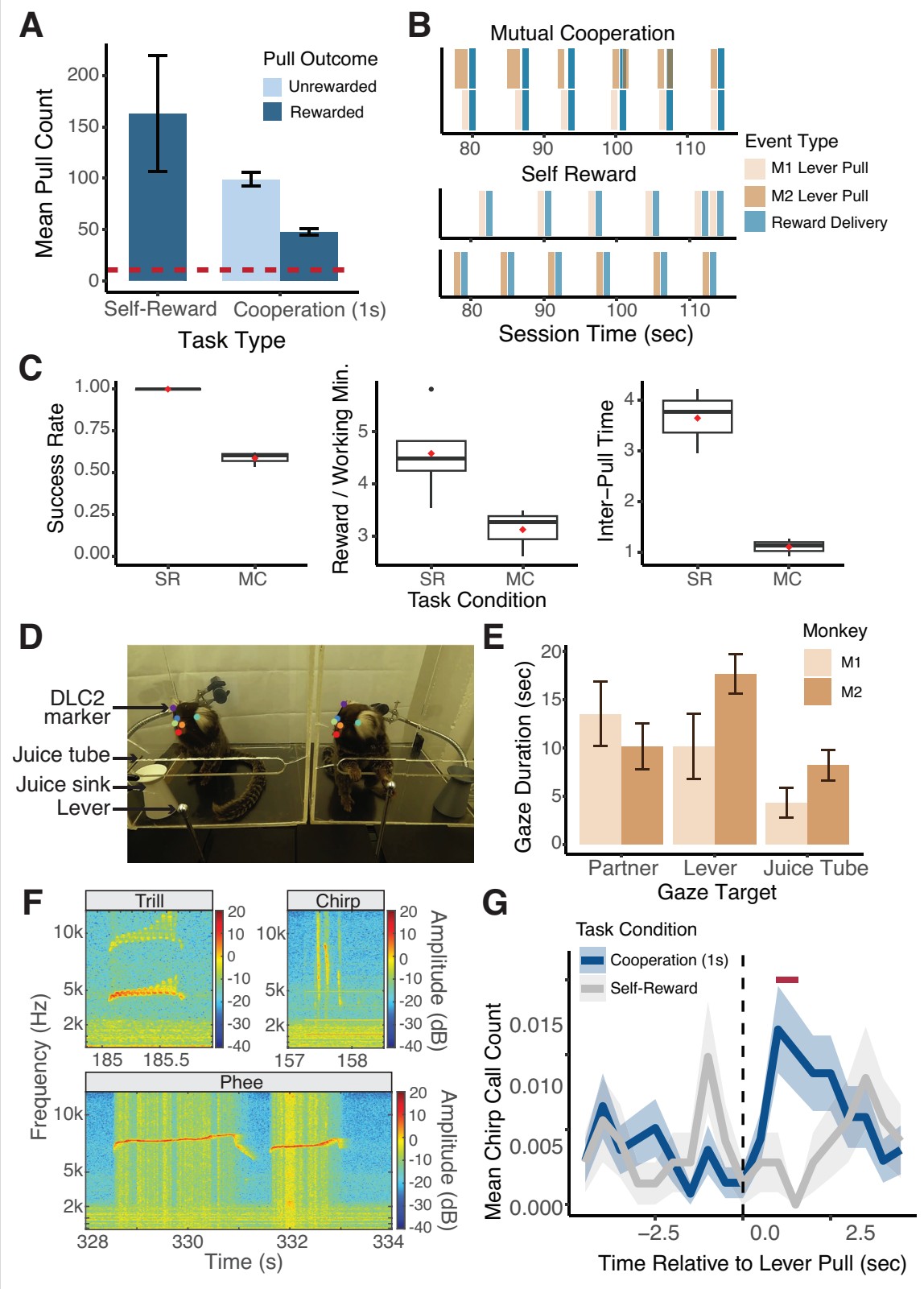

**Figure 2.** Quantitative behavioral measurements with Marmoset Apparatus for Automated Pulling (MarmoAAP). (**A**) Quantification of average number of lever pulls performed per animal in Self-Reward and 1 s Mutual Cooperation sessions (mean ± s.e.m.; n = 10 Self-Reward sessions, 67 Cooperation sessions). Red dashed line indicates the average number of trials collected per session from a sample of non-automated cooperative pulling paradigm experiments (***Cronin et al., 2005***; ***Martin et al., 2021***; ***Mendres, 2000***; ***Plotnik et al., 2011***; ***Range et al., 2019***; ***Drea and Carter, 2009***; ***Seed***

*Figure 2 continued on next page*

Figure 2 continued

et al., 2008). (**B**) Event time series example from a Mutual Cooperation and Self-Reward session. Each bar represents a task-related event (tan: monkey 1 lever pull, brown: monkey 2 lever pull, blue: reward delivery). (**C**) Metrics quantifying performance on MarmoAAP from six example sessions (three Self-Reward, three Mutual Cooperation). Red diamond indicates the mean. Left: Success rate calculated as number of successful lever pulls divided by number of total lever pulls in a session. Middle: Rewards earned per working minute calculated as the average number of rewards earned in 60 s for cumulative time in a session in which a monkey had pulled a lever within 30 s of that time. Right: Inter-pull time calculated as the average amount of time between M1 and M2 lever pulls in a session. (**D**) Example of DLC2 labeled video frame as marmosets perform cooperative pulling task. (**E**) Quantification of gaze targets averaged across three example cooperation sessions (mean ± s.e.m., n = 2 marmosets with 4 sessions each). (**F**) Example vocalizations captured during behavioral sessions. (**G**) Peristimulus time histogram (mean ± s.e.m.; n = 1103 Cooperation lever pulls, 566 Self-Reward lever pulls) of chirp vocalizations from one marmoset across Self-Reward (gray) and Mutual Cooperation (blue) sessions. Red bar indicate time bins with significantly different call counts for Cooperation compared to Self-Reward task conditions (Wilcoxon rank sum test, $p < 0.05$).

lever-pulling behavior, providing a dependable means to elicit a high number of motivated and appetitive behaviors.

Building upon this foundation, we extended our investigations to more complex tasks requiring cooperative pulling behaviors. We trained three unique dyads of familiar marmosets to perform cooperative pulling. For the Mutual Cooperation condition, we introduced a contingency that requires both marmosets to pull their levers within a specified time window of one another to both earn juice rewards. This task is completely unconstrained such that marmosets were free to pull their levers at any time. If the marmosets pulled their levers within the 1 s cooperative time window, a tone was played, and 0.2 ml of juice reward was delivered to both marmosets. However, if they pulled their lever and their partner did not pull within the cooperative time window, they were not rewarded.

We observed that all three dyads achieved proficiency, defined as 50% success rate (the number of pulls resulting in successful cooperation out of the total number of pulls made by both animals), in this cooperative task within a 2-month period. The choice of a 50% success criterion was based on pilot behavioral testing with marmosets, where this rate was deemed a reasonable target. Additionally, we referred to the findings of *Jiang et al., 2021*, who conducted a similar task with nose-poking behavior in mice, rats, and tree shrews. In their study, rats and tree shrews were observed to achieve and typically plateau around a 50% success rate, leading us to reason that marmosets could reach a similar level of performance. The average number of training days to reach Mutual Cooperation proficiency was 33.7±11.9 days.

This learning progression was further characterized by quantifiable metrics reflecting their progression in learning the cooperative contingency. Using the precise timestamps of the behavioral events (*Figure 2B*), we calculated three metrics across six example sessions (three Self-Reward sessions, three Mutual Cooperation sessions) that differentiated marmosets' performance on the Self-Reward and Mutual Cooperation tasks (*Figure 2C*). These included success rate, calculated as the number of successful lever pulls divided by the total number of lever pulls in a session; rewards earned per working minute, the average number of rewards earned per 60 s of active task engagement; and inter-pull time, the average time interval between lever pulls by the two marmosets (M1 and M2) in a session. We use these metrics here to demonstrate distinct patterns of performance between the Self-Reward and Mutual Cooperation tasks, highlighting specific behavioral markers associated with each task. Future work can utilize these metrics to track learning progress and define or quantify performance on this task under varying conditions.

Notably, the utilization of the automated behavioral paradigm enabled marmosets to perform an average of 146 trials per 20 min behavioral session across all sessions of learning the task, with an average of 47.9±3.3 trials (mean ± s.e.m.) resulting in successful level pulls (i.e. cooperation in Mutual Cooperation condition) and 98.9±6.6 trials resulting in unsuccessful attempts (*Figure 2A*). It's important to note that these numbers reflect performance across all sessions, including those during the initial learning phase, where success rates were often below the 50% proficiency threshold. This is a substantial improvement compared to the average of 10.4 trials per session observed in studies employing more manual cooperation paradigms. This heightened throughput, coupled with high repeatability, emerges as a critical asset for dissecting the intricacies of behavioral dynamics and the neural computations underlying complex behaviors.

## Customizable task parameters allow for adaptation to marmosets' abilities

Our behavior training also underscored the importance of task parameter adjustability in optimizing marmosets' performance. We tailored MarmoAAP to individual marmosets by fine-tuning parameters such as the distance required for a lever pull to register as a full pull and the force needed to initiate the lever-pulling action. By initially reducing the force required to pull the lever to 50 g, marmosets were able to smoothly transition into learning the task. Once they became habituated to the lever-pulling paradigm, we increased the lever-pulling force to 100 g and maintained this force level for Self-Reward and Mutual Cooperation tasks.

Additionally, we customized the reward magnitude offered for task completion to suit the specific requirements of the cooperative pulling task. For example, while marmosets exhibited motivation to work for a 0.1 ml juice reward in the individual pulling task, this was often not sufficient to elicit consistent pulling behaviors from dyads in the more difficult cooperative task. However, increasing the reward amount to 0.2 ml elicited enhanced motivation and more consistent cooperative behaviors from all three dyads. This adaptive parameter manipulation contributed significantly to the success of our marmoset dyads in mastering the cooperative pulling task, highlighting the importance of tailoring task parameters to individual and task-specific requirements.

Furthermore, MarmoAAP can easily be adapted to a wide variety of behavioral paradigms both in terms of the hardware configuration and parameters set by the task code. Given the modular nature of the apparatus design, the assembly can easily be adjusted to increase or decrease the number of pull levers as well as to change their configuration relative to one another. The task requirements imposed on the animals can also be easily adjusted by changing the task code. For example, the cooperative pull timing contingency, the force required to pull the lever, lever pull distance, and reward timing are just a few examples of task parameters that can be adjusted through the task code. One can imagine a wide variety of experiments that could be achieved with this apparatus to test cognitive processes such as, but not limited to, observational learning, memory, competition, altruism, executive function, and a host of other motivated behaviors.

## High-resolution behavioral data allows for advanced analyses

MarmoAAP facilitates comprehensive collection of detailed behavioral data across a variety of modalities. Its design allows for the capture of millisecond level outputs detailing lever positioning (*Figure 2B*) and the force applied to the lever. Additionally, it can be built to support cameras to record multiple angles for video data collection and incorporate microphones to record audio. Leveraging this video data, we used automated behavioral marking tools like DeepLabCut2 (DLC2) (*Mathis et al., 2018*; *Nath et al., 2019*) to obtain frame-by-frame annotations of the marmosets' head frames (*Figure 2D*). This rich dataset serves as a foundation for subsequent analyses, including the exploration of inferred gaze direction, spatial location within the enclosure, and overall movement trajectories.

In particular, we would like to highlight our ability to analyze gaze dynamics in this platform. Gaze behaviors are fundamental to social behaviors of primates which are highly visual animals. We were able to analyze complex behavioral dynamics by employing DLC2 to track the head frames of each freely moving marmoset as they engaged in the pulling task. We then used Anipose to create a 3D reconstruction of the marmosets' head frames based on videos from three cameras (*Karashchuk et al., 2021*). Based upon the constructed head frames, we estimated the marmosets' gaze direction by creating a virtual cone with an axis perpendicular to the plane defined by markers for the marmosets' eyes and forehead and a solid angle of 15 degrees. Using this approach, we were able to quantify the number of gazes toward various targets during the sessions. Using data from three example sessions, we can quantify bouts of gazes at targets of interest including their partner (social gaze), lever, and juice tube (*Figure 2E*). Such additional information that can be obtained within the automatic pulling paradigm can be used to better understand complex social interactions in marmoset pairs or groups.

Recognizing the highly vocal nature of marmosets and their extensive repertoire of vocalizations, each with distinct functions, we also collected audio recordings of every behavioral session. We were able to capture a wide variety of marmoset vocalizations during this task. Here, we specifically focused on chirp, trill, and phee calls (*Figure 2F*). Using the timestamps from the lever pulls and reward delivery, we further analyzed vocalizations relative to task events. As an example, we examined

vocalizations relative to successful and unsuccessful lever pulls from one marmoset across 19 sessions (6 Self-Reward sessions, 13 Mutual Cooperation sessions) (*Figure 2G*). This marmoset showed an increase in chirp calls, known to serve as food calls (*Rogers et al., 2018*; *Vitale et al., 2003*), after lever pulls in Mutual Cooperation sessions compared to Self-Reward sessions.

By incorporating these behavioral metrics from video and audio recordings with the timing of marmosets' pulling behaviors and reward delivery, one can gain a more comprehensive understanding of the intricate interplay between behavior, vocal communication, and cooperative interactions in this species using an automated pulling task.

## Precise synchronization with reproducible behavior allows behavior-locked neural data analyses

In addition to providing rich behavioral data and offering flexibility for various tasks, MarmoAAP and associated behavioral paradigms create an avenue for simultaneous neural recordings while freely moving marmosets are engaged in tasks implemented by MarmoAAP. MarmoAAP significantly increases the number of trials available for analysis and thus ensures ample statistical power when investigating the relationship between neural activity and behaviors. The highly reproducible lever-pulling behavior in marmosets within a naturalistic context strikes a crucial balance between conventional laboratory tests, where monkeys are immobilized and tasks lack natural movement but are tightly controlled, and more naturalistic animal behavior studies, where animals exhibit unrestrained behavior but lack regular behavioral benchmarks for studying the underlying neural dynamics (*Fan et al., 2021*; *Knöll et al., 2018*).

To validate this application of MarmoAAP, we conducted wireless neural recordings using a silicon-based linear array probe while a marmoset engaged in the cooperative pulling task with its partner (*Figure 3A*) and were able to isolate single-unit activity from the prefrontal cortex (*Figure 3B*). On each day, marmosets performed a 10 min session of the Mutual Cooperation task and a 10 min session of the Self-Reward task. By synchronizing the behavioral and neural activity timestamps, we were able to investigate spiking activity relative to various behavior events. Here, we present an example single unit recorded from the orbitofrontal cortex (OFC) and an example multi-unit from the dorsolateral prefrontal cortex (dlPFC) that showed increased firing rates around lever pulls in a Mutual Cooperation session (*Figure 3C and D*). Investigating neural activity with specific yet naturalistic behavioral events provides a valuable dataset for investigating the neural dynamics associated with cooperative interactions. By using wireless electrophysiology recording techniques in conjunction with this cooperative behavior paradigm with markerless behavioral tracking, one can obtain a more comprehensive understanding of the neural underpinnings of complex social behaviors, such as cooperation.

## Discussion

The Marmoset Automated Apparatus for Pulling (MarmoAAP) bridges the gap between traditional animal behavior methodologies and the demand for increased precision and adaptability in behavioral research. To advance our understanding of the complex behavioral and neural dynamics underlying cooperative behaviors, it is imperative that we transition toward a modernized approach to examining animal behaviors. In our current work, we introduced a novel automated cooperative pulling apparatus designed to address these limitations and advance the study of cooperative behaviors by providing a more refined and manipulable platform for experimentation. MarmoAAP offers the ability to enhance behavioral resolution in data collection, increase data output, streamline experimental procedures, and provide the flexibility to systematically manipulate task variables. With this scalable tool, researchers can gain insights into the behavioral dynamics governing cooperative behaviors and the neural mechanisms that underlie these complex social interactions. This methodology not only holds exceptional promise for enriching our understanding of primate behavior but also provides a unique opportunity to explore the intricate connections between neural processes and actions in a manner that bridges controlled and naturalistic experimental conditions.

The development of MarmoAAP arrives at a critical time, coinciding with burgeoning efforts to engineer genetically modified marmosets (*Kaiser and Feng, 2015*; *Kishi et al., 2014*; *Kumita et al., 2019*; *Sato et al., 2016*). As such models progress, it is essential to have robust methodologies that can accurately measure the features of marmoset social interactions. Precise behavioral assays are

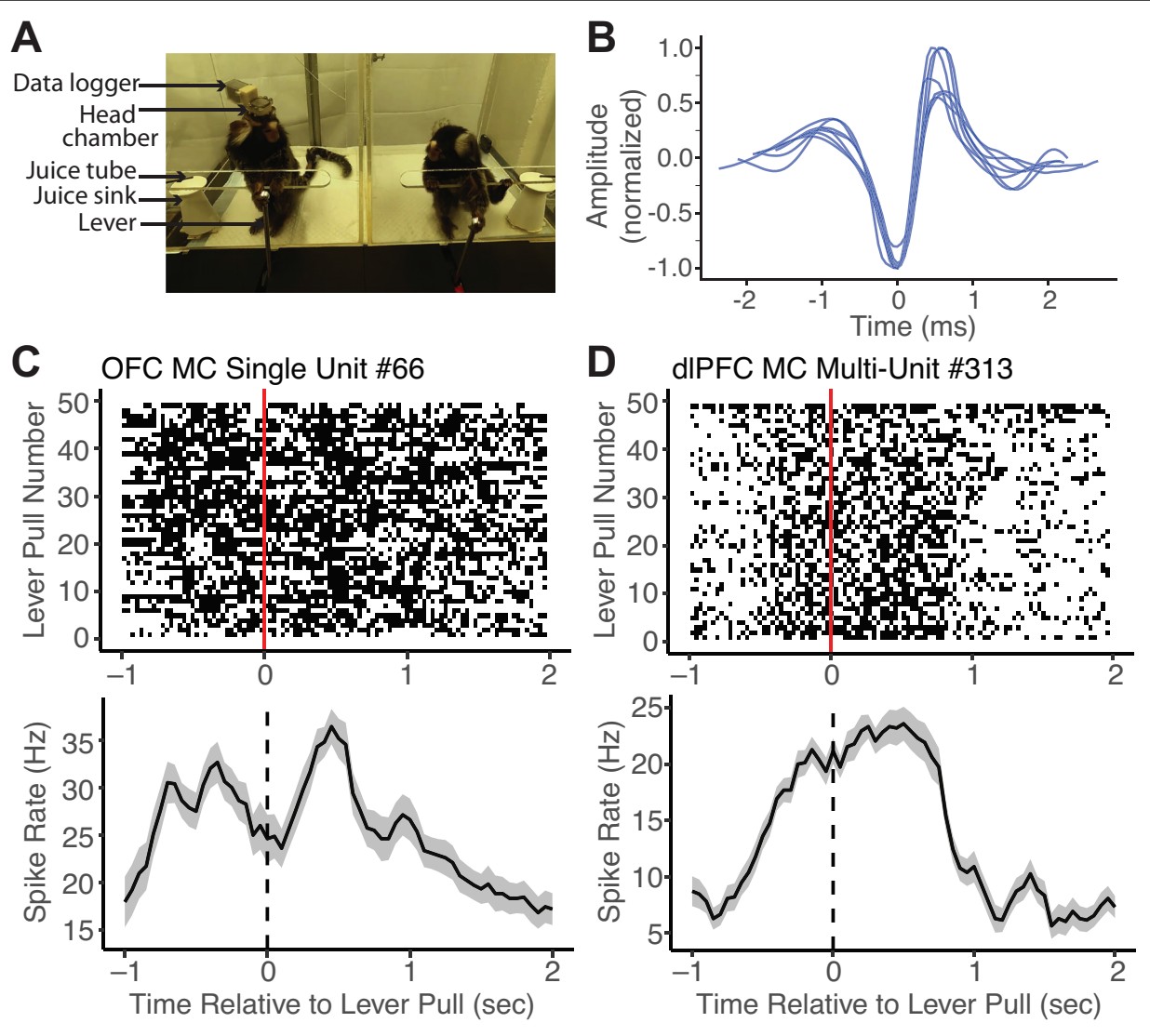

**Figure 3.** Wireless neural data recordings with Marmoset Apparatus for Automated Pulling (MarmoAAP). (**A**) Photo of marmosets performing Mutual Cooperation task with one marmoset performing the task with a head-mounted wireless recording system. (**B**) Example single-unit waveforms from one recording session of the orbitofrontal cortex (OFC). (**C**) Top: Peristimulus time histogram of an example single OFC neuron averaged across all self lever pulls in one session aligned to the time of lever pull registration (dashed line) (n = 49 lever pulls, bin size = 150 ms with 50 ms sliding window). Bottom: Raster plot of the same example OFC neuron relative to all self lever pulls (red line = lever pull registration) in one example session. (**D**) Top: Peristimulus time histogram of an example dorsolateral prefrontal cortex (dlPFC) multi-unit averaged across all self lever pulls in one session aligned to the time of lever pull registration (dashed line) (n = 49 lever pulls, bin size = 150 ms with 50 ms sliding window). Bottom: Raster plot of example dlPFC multi-unit relative to all self lever pulls (red line = lever pull registration) in one example session.

indispensable for future investigations aiming to elucidate the effects of genetic modifications on social behavior and test potential therapeutic approaches. Just as neurological abilities such as locomotion can be quantitatively assessed (*Pickett et al., 2020*), it is critical to establish equivalent metrics for evaluating complex behavioral patterns in marmosets.

## Automated task paradigm for naturalistic social exploration

Using MarmoAAP, we were able to elicit consistent and highly repeatable motivated behaviors in freely moving marmoset monkeys. This task design strikes a pivotal balance between traditional naturalistic animal behavior studies, which benefit from a high degree of naturalism but often suffer from low behavioral resolution and limited trial counts, and conventional lab studies, which are highly controlled but lack natural ethological relevance (*Fan et al., 2021*). Previous research has underscored

the substantial impact of behavioral context, specifically the distinction between constrained and freely moving conditions, on prefrontal cortical representations of social information (*Jovanovic et al., 2022*). Our shift toward a paradigm that integrates naturalistic, yet highly repeatable, decisions and actions is imperative for the comprehensive exploration of natural social behaviors and the elucidation of their underlying neural mechanisms. This approach addresses the limitations of paradigms that fall short of faithfully capturing the intricacies of social interactions, emphasizing the importance of a more ecologically valid framework for advancing our understanding of the neural dynamics that underpin fundamental aspects of primate social brain functions.

## Quantification of high-throughput cooperative behaviors

We show that marmosets exhibit a rapid acquisition of proficiency in the lever-pulling action and demonstrate their capacity to grasp more complex task contingencies, such as the cooperative pulling task highlighted in this study. Our findings also showcase that the detailed behavioral data outputs from the apparatus, including millisecond-level timestamps for lever pulls and reward deliveries, enable us to quantitatively assess marmosets' learning and performance on this task. Significantly, our study demonstrates that the automated apparatus facilitates a substantial 15-fold increase in the number of trials conducted per session compared to conventional pulling paradigms. This increased trial throughput is of critical importance for investigations of the neural mechanisms underlying these social behaviors, ensuring the acquisition of a robust dataset for comprehensive analyses of neural activity during naturalistic behavioral settings. Moreover, the ability to examine complex social interactions with high-throughput data might be particularly important for characterizing transgenic marmoset models.

While our study demonstrates a considerable increase in trial throughput using the automated apparatus, with marmosets completing an average of 146 trials per 20 min session, it is important to note that not all of these trials result in successful outcomes. On average, 47.9 of the 146 trials were successful. Additionally, there are challenges associated with sustaining motivation over longer periods, particularly in the context of the Mutual Cooperation task which requires not just one, but two animals to be simultaneously motivated and engaged for the same duration. Coordinating motivation between two animals is inherently more challenging than motivating a single subject, as both must be willing to work at the same time and for the same length of time to achieve successful cooperation.

We experimented with several liquid rewards and ultimately selected diluted marshmallow water, as it was consistently consumed by all marmosets. However, further optimization tailored to individual marmosets' preferences could potentially enhance motivation and extend the duration of task engagement, thereby increasing the number of trials per session. Additionally, our approach to food and water restriction was minimal, involving only the removal of food and water for 1–3 hr each morning without limiting the overall daily intake. For future studies, researchers might consider implementing more controlled and stringent food and water restrictions, in line with established protocols, to increase the marmosets' motivation by ensuring they are sufficiently hungry or thirsty during task sessions. While our primary focus was on the design, development, and validation of the apparatus and methods, we recognize the potential for further optimization in these areas to maximize the efficacy of the paradigm for neurophysiological and cognitive experiments.

## Manipulability and adaptability of task parameters and apparatus

Importantly, the configuration of MarmoAAP allows for precise adjustment of task parameters, a key feature for optimizing marmoset performance and facilitating investigations into a diverse array of complex behaviors. Experimenters can easily fine-tune parameters in the task code to customize apparatus functionality for various behavioral tasks or to accommodate the specific needs and capabilities of individual animals. This adaptability not only expedites the animal training process but also allows for a nuanced exploration of the intricate dimensions of cooperative behaviors, ensuring that experimental conditions closely align with research objectives. Additionally, the design of MarmoAAP is modular, enabling it to be built in different configurations, such as varying the positioning or number of motors and levers. This modularity allows the apparatus to be adapted not only for different types of social tasks but also for non-social tasks, further expanding its utility. Such flexibility is indispensable not only for cooperative tasks but also positions our paradigm as a versatile tool for delving into cognitive processes beyond cooperation.

Although this paradigm was developed specifically for marmosets, its adaptable design suggests it could be readily modified for use in other species. One key modification would involve adjusting the size of the servo motor and the lever, as the current setup is tailored to small animals like marmosets, which can be trained to exert a pulling force of approximately 500–600 g. For larger animals, incorporating a larger motor capable of exerting greater force, along with more durable parts, would be recommended. By making these adjustments, researchers could tailor the paradigm to suit the physical and cognitive characteristics of different animals, enabling comparative studies across species. This is particularly valuable, as it allows for the examination of how various species approach the same social challenges, providing deeper insights into the nuances of their socio-cognitive abilities.

### High-resolution behavioral data and multimodal analyses

In tandem with the intricate behavioral outputs derived from the apparatus, MarmoAAP incorporates the integration of information from many sources and modalities. Utilizing video recordings obtained during the task, we showcased the application of automated behavioral marking tools, such as DLC2 (*Mathis et al., 2018*; *Nath et al., 2019*), to probe the interplay between behavioral dynamics—particularly gaze behaviors—and performance on the cooperative task. Complementarily, the inclusion of audio recordings enriches this dataset, allowing for a comprehensive examination of marmosets' vocal communication patterns and their correlation with task events. This multimodal approach establishes a robust foundation for nuanced investigations into the cognitive processes and social dynamics of marmosets, aligning with a goal toward a comprehensive understanding of primate social behaviors.

### Integration with neural recordings

A key attribute of the MarmoAAP design is its capacity to seamlessly integrate with wireless electrophysiology recordings, providing an avenue to explore the neural underpinnings of behavioral processes. The apparatus allows for precise time-locking of task and behavioral events with neural activity as demonstrated in the dlPFC and the OFC. With a substantially increased number of trials amassed through MarmoAAP, this demonstration supports the possibility of examining the neural dynamics underlying cooperative behaviors in marmosets. Our apparatus and paradigm represent a noteworthy advancement, bridging the gap between traditional animal behavior studies that address ethologically relevant behaviors of animals and precise, highly controlled investigations of neural activity.

### Conclusion and future directions

In conclusion, we hope that MarmoAAP and the associated automated cooperative pulling paradigm will make a significant contribution to the study of marmoset social behaviors in the field. The combination of a highly modular and adaptable design, high-resolution behavioral data, and integration with neural recordings positions our paradigm as a robust and versatile tool for unraveling the complexities of primate behavior. As we move forward, this paradigm not only serves as a platform for in-depth investigations into marmoset social dynamics but also holds the promise of extending our understanding of cognitive processes and neural mechanisms across a variety of complex behaviors. The scientific community can leverage this paradigm to explore a myriad of cognitive processes, from observational learning to executive function, laying the groundwork for comprehensive insights into the neural mechanisms of complex behaviors in nonhuman primates.

## Acknowledgements

This work was supported by the National Science Foundation Graduate Research Fellowship (DGE2139841, OCM), the National Institute of Mental Health (R21 MH126072, SWCC, ASN, MPJ), the Simons Foundation Autism Research Initiative (SFARI 875855, SWCC, ASN, MPJ), Wu Tsai Institute at Yale University (SWCC, ASN, MPJ, WS), and a National Eye Institute core grant for vision research (P30 EY026878 to Yale University). We thank Paul Shamble and the Neurotechnology Core of the Kavli Institute for Neuroscience at Yale University for providing technical support. We also like to thank Feng Xing, Amrita Nair, and Nyomi Hudson for their support in this research project.

# Additional information

### Funding

| Funder | Grant reference number | Author |
|---|---|---|
| National Science Foundation | DGE2139841 | Olivia C Meisner |
| National Institute of Mental Health | R21 MH126072 | Monika P Jadi<br>Anirvan S Nandy<br>Steve WC Chang |
| Simons Foundation Autism Research Initiative | SFARI 875855 | Monika P Jadi<br>Anirvan S Nandy<br>Steve WC Chang |
| National Eye Institute | P30 EY026878 | Anirvan S Nandy |
| Wu Tsai Institute, Yale University | | Weikang Shi<br>Monika P Jadi<br>Anirvan S Nandy<br>Steve WC Chang |

The funders had no role in study design, data collection and interpretation, or the decision to submit the work for publication.

### Author contributions

Olivia C Meisner, Conceptualization, Data curation, Formal analysis, Investigation, Visualization, Methodology, Writing – original draft, Writing – review and editing; Weikang Shi, Data curation, Formal analysis, Investigation, Visualization, Methodology, Writing – original draft, Writing – review and editing; Nicholas A Fagan, Joel Greenwood, Software, Methodology; Monika P Jadi, Resources, Data curation, Formal analysis, Supervision, Funding acquisition, Investigation, Visualization, Methodology, Writing – original draft, Project administration, Writing – review and editing; Anirvan S Nandy, Steve WC Chang, Conceptualization, Resources, Data curation, Formal analysis, Supervision, Funding acquisition, Investigation, Visualization, Methodology, Writing – original draft, Project administration, Writing – review and editing

### Author ORCIDs

Olivia C Meisner ⓘ https://orcid.org/0000-0003-3523-5144
Weikang Shi ⓘ https://orcid.org/0000-0002-4068-1168
Monika P Jadi ⓘ https://orcid.org/0000-0003-1092-5026
Anirvan S Nandy ⓘ https://orcid.org/0000-0002-4225-5349
Steve WC Chang ⓘ https://orcid.org/0000-0003-4160-7549

### Ethics

All procedures were approved by the Yale Institutional Animal Care and Use Committee and complied with the National Institutes of Health Guide for the Care and Use of Laboratory Animals. (Yale University IACUC Protocol #2023-20163).

Reviewer #1 (Public Review): https://doi.org/10.7554/eLife.97088.3.sa1
Reviewer #2 (Public Review): https://doi.org/10.7554/eLife.97088.3.sa2
Reviewer #3 (Public Review): https://doi.org/10.7554/eLife.97088.3.sa3
Author response https://doi.org/10.7554/eLife.97088.3.sa4

# Additional files

### Supplementary files

• MDAR checklist

## Data availability

Code for the automated pulling tasks and the SolidWorks CAD file for the apparatus can be found at GitHub.

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
