## [Editor Report · eLife Assessment]

This **valuable** study describes an apparatus, workflow, and proof-of-concept data for a system to study social cooperation in marmosets, an increasingly popular primate model for neuroscience. The apparatus and methodology have clear and **convincing** advantages over conventional methods based on manual approaches. However, claims of faster social learning or of finer-grained behavioural analysis in this setup will require further corroboration.

---

## [Referee Report · Reviewer #1 (Public Review)]

Summary:

This manuscript by Meissner and colleagues described a novel take on a classic social cognition paradigm developed for marmosets. The classic pull task is a powerful paradigm that has been used for many years across numerous species, but its analog approach has several key limitations. As such, it has not been feasible to adopt the task for neuroscience experiments. Here the authors capture the spirit of the classic task but provide several fundamental innovations that modernize the paradigm - technically and conceptually. By developing the paradigm for marmosets, the authors leverage the many advantages of this primate model for studies of social brain functions and their particular amenability to freely-moving naturalistic approaches.

Strengths:

The current manuscript describes one of the most exciting paradigms in primate social cognition to be developed in many years. By allowing for freely-moving marmosets to engage in high numbers of trials, while precisely quantifying their visual behavior (e.g. gaze) and recording neural activity this paradigm has the potential to usher in a new wave of research on the cognitive and neural mechanisms underlying primate social cognition and decision-making. This paradigm is an elegant illustration of how naturalistic questions can be adapted to more rigorous experimental paradigms. Overall, I thought the manuscript was well written and provided sufficient details for others to adopt this paradigm.

---

## [Referee Report · Reviewer #2 (Public Review)]

Summary:

This important work by Meisner et al., developed an automated apparatus (MarmoAPP) to collect a wide array of behavioral data (lever pulling, gaze direction, vocalizations) in marmoset monkeys, with the goal of modernizing collection of behavioral data to coincide with the investigation of neurological mechanisms governing behavioral decision making in an important primate neuroscience model. The authors show a variety of "proof-of-principle" concepts that this apparatus can collect a wide range of behavioral data, with higher behavioral resolution than traditional methods. For example, the authors highlight that typical behavioral experiments on primate cooperation provide around 10 trials per session, while using their approach the authors were able to collect over 100 trials per 20-minute session with the MarmoAAP.

Overall the authors argue that this approach has a few notable advantages:

(1) It enhances behavioral output which is important for measuring small or nuanced effects/changes in behavior;

(2) Allows for more advanced analyses given the higher number of trials per session;

(3) Significantly reduces the human labor of manually coding behavioral outcomes and experimenter interventions such as reloading apparatuses for food or position;

(4) Allows for more flexibility and experimental rigor in measuring behavior and neural activity simultaneously.

Strengths:

The paper is well-written and the MarmoAPP appears to be highly successful at integrating behavioral data across many important contexts (cooperation, gaze, vocalizations), with the ability to measure significantly many more behavioral contexts (many of which the authors make suggestions for).

The authors provide substantive information about the design of the apparatus, how the apparatus can be obtained via a long list of information Apparatus parts and information, and provide data outcomes from a wide number of behavioral and neurological outcomes. The significance of the findings is important for the field of social neuroscience and the strength of evidence is solid in terms of the ability of the apparatus to perform as described, at least in marmoset monkeys. The advantage of collecting neural and freely-behaving behavioral data concurrently is a significant advantage.

---

## [Referee Report · Reviewer #3 (Public Review)]

Summary:

The authors set out to devise a system for the neural and behavioral study of socially cooperative behaviors in nonhuman primates (common marmosets). They describe instrumentation to allow for a "cooperative pulling" paradigm, the training process, and how both behavioral and neural data can be collected and analyzed. This is a valuable approach to an important topic, as the marmoset stands as a great platform to study primate social cognition. Given that the goals of such a methods paper are to (a) describe the approach and instrumentation, (b) show the feasibility of use, and (c) quantitatively compare to related approaches, the work is easily able to meet those criteria. My specific feedback on both strengths and weaknesses is therefore relatively limited in scope and depth.

Strengths:

The device is well-described, and the authors should be commended for their efforts in both designing this system but also in "writing it up" so that others can benefit from their R&D.

The device appears to generate more repetitions of key behavior than other approaches used in prior work (with other species).

The device allows for quantitative control and adjustment to control behaviour.

The approach also supports the integration of markerless behavioral analysis as well as neurophysiological data.

---

## [Author Response]

The following is the authors’ response to the original reviews.

**Reviewer #1 (Public Review):**
Summary:This manuscript by Meissner and colleagues described a novel take on a classic social cognition paradigm developed for marmosets. The classic pull task is a powerful paradigm that has been used for many years across numerous species, but its analog approach has several key limitations. As such, it has not been feasible to adopt the task for neuroscience experiments. Here the authors capture the spirit of the classic task but provide several fundamental innovations that modernize the paradigm - technically and conceptually. By developing the paradigm for marmosets, the authors leverage the many advantages of this primate model for studies of social brain functions and their particular amenability to freely-moving naturalistic approaches.Strengths:The current manuscript describes one of the most exciting paradigms in primate social cognition to be developed in many years. By allowing for freely-moving marmosets to engage in high numbers of trials, while precisely quantifying their visual behavior (e.g. gaze) and recording neural activity this paradigm has the potential to usher in a new wave of research on the cognitive and neural mechanisms underlying primate social cognition and decision-making. This paradigm is an elegant illustration of how naturalistic questions can be adapted to more rigorous experimental paradigms. Overall, I thought the manuscript was well written and provided sufficient details for others to adopt this paradigm. I did have a handful of questions and requests about topics and information that could help to further accelerate its adoption across the field.Weaknesses:LN 107 - Otters have also been successful at the classic pull task (https://link.springer.com/article/10.1007/s10071-017-1126-2)

We have added this reference to the manuscript.

LN 151 - Can you provide a more precise quantification of timing accuracy than the 'sub-second level'. This helps determine synchronization with other devices.

We have included more precise timing details, noting that data is stored at the millisecond level.

Using this paradigm, the marmosets achieved more trials than in the conventional task (146 vs 10). While this is impressive, given that only ~50 are successful Mutual Cooperation trials it does present some challenges for potential neurophysiology experiments and particular cognitive questions. The marmosets are only performing the task for 20 minutes, presumably because they become sated and are no longer motivated. This seems a limitation of the task and is something worth discussing in the manuscript. Did the authors try other food rewards, reduce the amount of reward, food/water restrict the animals for more than the stated 1-3 hours? How might this paradigm be incorporated into in-cage approaches that have been successful in marmosets? Any details on this would help guide others seeking to extend the number of trials performed each day.

We have added a discussion addressing the use of liquid rewards, minimal food and water restriction, and the potential for further optimization to increase task engagement and trial numbers. This is now reflected in the revised manuscript.

Can you provide more details on the DLC/Anipose procedure? How were the cameras synchronized? What percentage of trials needed to be annotated before the model could be generalized? Did each monkey require its own model, or was a single one applied to all animals?

We have added more detailed information on the DLC and Anipose tracking which can be found in the *Multi-animal 3D tracking* section under Materials & Methods.

Will the schematics and more instructions on building this system be made publicly available? A number of the components listed in Table 1 are custom-designed. Although it is stated that CAD files will be made available upon request, sharing a link to these files in an accessible folder would significantly add to the potential impact of this paradigm by making it easier for others to adopt.

We have made the SolidWorks CAD files publicly available. They can now be found in the Github repository alongside the apparatus and task code.

In the Discussion, it would be helpful to have some discussion of how this paradigm might be used more broadly. The classic pulling paradigm typically allows one to ask a specific question about social cognition, but this task has the potential to be more widely applied to other social decision-making questions. For example, how might this task be adopted to ask some of the game-theory-type approaches common in this literature? Given the authors' expertise in this area, this discussion could serve to provide a roadmap for the broader field to adopt.Although this paradigm was developed specifically for marmosets, it seems to me that it could readily be adopted in other species with some modifications. Could the authors speak to this and their thoughts on what may need to be changed to be used in other species? This is particularly important because one of the advantages of the classic paradigm is that it has been used in so many species, providing the opportunity to compare how different species approach the same challenge. For example, though both chimps and bonobos are successful, their differences are notably illuminating about the nuances of their respective social cognitive faculties.

We have expanded the discussion for the broader applications of this apparatus both for other decision-making research questions as well as its adaptability for use in other species.

**Reviewer #2 (Public Review):**
Summary:This important work by Meisner et al., developed an automated apparatus (MarmoAPP) to collect a wide array of behavioral data (lever pulling, gaze direction, vocalizations) in marmoset monkeys, with the goal of modernizing collection of behavioral data to coincide with the investigation of neurological mechanisms governing behavioral decision making in an important primate neuroscience model. The authors show a variety of "proof-of-principle" concepts that this apparatus can collect a wide range of behavioral data, with higher behavioral resolution than traditional methods. For example, the authors highlight that typical behavioral experiments on primate cooperation provide around 10 trials per session, while using their approach the authors were able to collect over 100 trials per 20-minute session with the MarmoAAP.Overall the authors argue that this approach has a few notable advantages:(1) it enhances behavioral output which is important for measuring small or nuanced effects/changes in behavior;(2) allows for more advanced analyses given the higher number of trials per session;(3) significantly reduces the human labor of manually coding behavioral outcomes and experimenter interventions such as reloading apparatuses for food or position;(4) allows for more flexibility and experimental rigor in measuring behavior and neural activity simultaneously.Strengths:The paper is well-written and the MarmoAPP appears to be highly successful at integrating behavioral data across many important contexts (cooperation, gaze, vocalizations), with the ability to measure significantly many more behavioral contexts (many of which the authors make suggestions for).The authors provide substantive information about the design of the apparatus, how the apparatus can be obtained via a long list of information Apparatus parts and information, and provide data outcomes from a wide number of behavioral and neurological outcomes. The significance of the findings is important for the field of social neuroscience and the strength of evidence is solid in terms of the ability of the apparatus to perform as described, at least in marmoset monkeys. The advantage of collecting neural and freely-behaving behavioral data concurrently is a significant advantage.Weaknesses:While this paper has many significant strengths, there are a few notable weaknesses in that many of the advantages are not explicitly demonstrated within the evidence presented in the paper. There are data reported (as shown in Figures 2 and 3), but in many cases, it is unclear if the data is referenced in other published work, as the data analysis is not described and/or self-contained within the manuscript, which it should be for readers to understand the nature of the data shown in Figures 2 and 3.(1) There is no data in the paper or reference demonstrating training performance in the marmosets. For example, how many sessions are required to reach a pre-determined criterion of acceptable demonstration of task competence? The authors reference reliably performing the self-reward task, but this was not objectively stated in terms of what level of reliability was used. Moreover, in the Mutual Cooperation paradigm, while there is data reported on performance between self-reward vs mutual cooperation tasks, it is unclear how the authors measured individual understanding of mutual cooperation in this paradigm (cooperation performance in the mutual cooperation paradigm in the presence or absence of a partner; and how, if at all, this performance varied across social context). What positive or negative control is used to discern gained advantages between deliberate cooperation vs two individuals succeeding at self-reward simultaneously?

Thank you for your comment. This Tools & Resources paper is focused solely on the development of the apparatus and methods. Future publications will provide more details on training performance, learning behaviors, and include appropriate controls to distinguish deliberate cooperation from simultaneous success in self-reward tasks.

(2) One of the notable strengths of this approach argued by the authors is the improved ability to utilize trials for data analysis, but this is not presented or supported in the manuscript. For example, the paper would be improved by explicitly showing a significant improvement in the analytical outcome associated with a comparison of cooperation performance in the context of ~150 trials using MarmoAAP vs 10-12 trials using conventional behavioral approaches beyond the general principle of sample size. The authors highlight the dissection of intricacies of behavioral dynamics, but more could be demonstrated to specifically show these intricacies compared to conventional approaches. Given the cost and expertise required to build and operate the MarmoAAP, it is critical to provide an important advantage gained on this front. The addition of data analysis and explicit description(s) of other analytical advantages would likely strengthen this paper and the advantages of MarmoAAP over other behavioral techniques.

Thank you for the suggestion. While this manuscript focuses on the apparatus and methods, the increase in trial numbers itself provides clear advantages, including greater statistical power and more robust analyses of behavioral dynamics. Future publications will offer more in-depth analyses comparing the performance and cooperation behavior observed with MarmoAAP, further demonstrating these analytical benefits.

**Reviewer #3 (Public Review):**
Summary:The authors set out to devise a system for the neural and behavioral study of socially cooperative behaviors in nonhuman primates (common marmosets). They describe instrumentation to allow for a "cooperative pulling" paradigm, the training process, and how both behavioral and neural data can be collected and analyzed. This is a valuable approach to an important topic, as the marmoset stands as a great platform to study primate social cognition. Given that the goals of such a methods paper are to (a) describe the approach and instrumentation, (b) show the feasibility of use, and (c) quantitatively compare to related approaches, the work is easily able to meet those criteria. My specific feedback on both strengths and weaknesses is therefore relatively limited in scope and depth.Strengths:The device is well-described, and the authors should be commended for their efforts in both designing this system but also in "writing it up" so that others can benefit from their R&D.The device appears to generate more repetitions of key behavior than other approaches used in prior work (with other species).The device allows for quantitative control and adjustment to control behavior.The approach also supports the integration of markerless behavioral analysis as well as neurophysiological data.Weaknesses:A few ambiguities in the descriptions are flagged below in the "Recommendations for authors".The system is well-suited to marmosets, but it is less clear whether it could be generalized for use in other species (in which similar behaviors have been studied with far less elegant approaches). If the system could impact work in other species, the scope of impact would be significantly increased, and would also allow for more direct cross-species comparisons. Regardless, the future work that this system will allow in the marmoset will itself be novel, unique, and likely to support major insights into primate social cognition.

Thank you for this feedback. We have expanded the discussion to include how the apparatus could be adapted for use in other species, highlighting the potential modifications required, such as adjusting the size and strength of the servo motor and components. These changes would enable broader applications and facilitate cross-species comparisons.